# Evaluating Landscape Attractiveness with Geospatial Data, A Case Study in Flanders, Belgium

**Astrid Vannoppen** [1,*], **Jeroen Degerickx** [1] **and Anne Gobin** [1,2]

1   Vlaamse Instelling voor Technologisch Onderzoek NV, 2400 Mol, Belgium; jeroen.degerickx@vito.be (J.D.); anne.gobin@kuleuven.be (A.G.)
2   Department of Earth and Environmental Sciences, Faculty of BioScience Engineering, University of Leuven, 3001 Leuven, Belgium
*   Correspondence: astrid.vannoppen@vito.be

**Abstract:** Attractive landscapes are diverse and resilient landscapes that provide a multitude of essential ecosystem services. The development of landscape policy to protect and improve landscape attractiveness, thereby ensuring the provision of ecosystem services, is ideally adapted to region specific landscape characteristics. In addition, trends in landscape attractiveness may be linked to certain policies, or the absence of policies over time. A spatial and temporal evaluation of landscape attractiveness is thus desirable for landscape policy development. In this paper, landscape attractiveness was spatially evaluated for Flanders (Belgium) using landscape indicators derived from geospatial data as a case study. Large local differences in landscape quality in (i) rural versus urban areas and (ii) between the seven agricultural regions in Flanders were found. This observed spatial variability in landscape attractiveness demonstrated that a localized approach, considering the geophysical characteristics of each individual region, would be required in the development of landscape policy to improve landscape quality in Flanders. Some trends in landscape attractiveness were related to agriculture in Flanders, e.g., a slight decrease in total agricultural area, decrease in dominance of grassland, maize and cereals, a decrease in crop diversity, sharp increase in the adoption of agri-environmental agreements (AEA) and a decrease in bare soil conditions in winter. The observed trends and spatial variation in landscape attractiveness can be used as a tool to support policy analysis, assess the potential effects of future policy plans, identify policy gaps and evaluate past landscape policy.

**Keywords:** trend analysis; landscape quality indicators; land cover; biodiversity; agricultural management; GIS; remote sensing; landscape attractiveness



## 1. Introduction

An attractive landscape is more than merely a visually appealing landscape. It is defined as a resilient and diverse landscape providing a multitude of ecosystem services and playing a key role in overcoming some of the pressing challenges of the 21st century such as climate change and urbanization [1–3]. The European Landscape Convention therefore encourages EU Member States to monitor, protect and improve the attractiveness of the European landscape [1]. Monitoring and policy development to protect and improve landscape attractiveness should go hand in hand [1]. A monitoring framework that can easily be adapted to regional characteristics and data availability is therefore desirable. Qualitative data on landscape attractiveness can be very useful in foresight exercises, which are often used in future (rural) policy development [4].

Landscape attractiveness can be monitored with a multitude of methods. Generally, only the visual landscape is used to evaluate landscape attractiveness [2]. For example, landscape attractiveness can be assessed by analyzing the public and stakeholder perceptions of the landscape using participatory methods such as interviews and questionnaires [5–7]. Widely available geospatial data of the visual landscape are often used to

assess landscape attractiveness [2,8,9]. In case of the latter, the large amount of data that characterize the landscape can be translated into qualitative and quantitative landscape indicators, which allow an objective evaluation of the landscape [2,10]. In addition, geospatial data are often available for multiple points in time, allowing the assessment of trends and changes in landscape attractiveness. The quality of the visual landscape layer is the result of the combined effect of its spatial structure, ecological properties (e.g., ecosystem functioning), visual features (e.g., harmony of forms) and cultural value (e.g., state of the cultural heritage) [2,11]. For each of these landscape dimensions, geospatial data can be used to calculate landscape indicators.

So-called landscape metrics are landscape indicators used to describe the structural dimension of the landscape. Examples of landscape metrics are indicators describing the area, shape, adjacency and fragmentation of landscape units. Specific software exists to calculate these types of landscape indicators [12]. Common landscape metrics that are used to evaluate the structural landscape dimension are the number and shape of landscape patches and the diversity and fragmentation of the landscape [2,13]. These landscape metrics can be easily calculated from geospatial land cover maps [2]. Assessing the visual dimension of the landscape with geospatially derived indicators is challenging as this is a function of personal preferences (i.e., cultural background) and the conditions of assessment (i.e., weather conditions) [2]. The ecological dimension of the landscape can be assessed by geospatially derived landscape indicators describing the functioning, stability, biodiversity and fragmentation of ecosystems [14]. Additionally, [15,16] demonstrated that remotely sensed spectral data and vegetation indices, such as NDVI, can be used to evaluate tree species diversity. There are a multitude of landscape indicators describing the cultural dimension of the landscape (see [17]). However, only a few of these indicators, such as the existence of "old habitat trees" [18], can be calculated based on geospatial data [2].

The translation of landscape indicators into landscape attractiveness is commonly achieved by combining expert knowledge and public opinion. Landscape has, however, also a non-visual layer that reflects the social and economic dimensions, which is not easily analyzed using geospatial data. Data on the population density, willingness to pay to maintain or restore a certain landscape and the cost of conservation of a specific landscape type are examples of indicators of the social and economic dimensions of the landscape [2].

The aim of this paper is to evaluate landscape attractiveness based on landscape indicators derived from geospatial data and establish an integrated landscape attractiveness score. Similar to [2,11,19], geospatial landscape indicators describing different dimensions of the landscape were used to evaluate landscape attractiveness. We applied the methodology to the region of Flanders in Belgium at different spatial scales. Flanders has one of the highest population densities in Europe, i.e., 484 inhabitants per square kilometer in 2019, greatly exceeding the European average of 118.6 inhabitants per square kilometer in 2018 [20,21]. The high population density, in combination with poor spatial planning in Flanders, resulted in a landscape that is intensively used, fragmentated, has a complex spatial structure and is highly heterogeneous [22]. Landscape indicators were analyzed spatially and temporally (depending on data availability) to comprehend this complex landscape and its recent evolution. The spatial evaluation of landscape indicators was performed for (i) Flanders, (ii) rural versus urban areas and (iii) the seven agricultural regions of Flanders. The landscape indicators were integrated for each studied spatial unit into a landscape attractiveness score.

## 2. Materials and Methods

### 2.1. Landscape Attractiveness Monitoring Framework

Landscape attractiveness was assessed based on three key landscape quality aspects: structural, ecological and management quality. For each of these landscape quality aspects a set of indicators derived from available geospatial databases and remote sensing data was used. In a final stage, the landscape quality indicators were integrated to obtain an overall

score of landscape attractiveness. The following sections present (i) an overview of the indicators used to assess structural, ecological and management landscape quality aspects and (ii) the methodology to integrate these into an overall landscape attractiveness score.

### 2.1.1. Landscape Attractiveness Indicators

Landscape attractiveness includes structural, ecological, visual and cultural dimensions [2,11]. In this paper, we decided to discard the cultural dimension as landscape indicators commonly used to assess historical monuments and historical landscape elements (see for example [2]) can be regarded as static over a short time frame. In addition, we did not include purely visual aspects such as form, color and shape disharmony indices, as suggested by [2], that relate strongly to mental health [23]. Instead, we focused on aspects of the landscape with a clear linkage to ecosystem service provisioning, i.e., the structural appearance of the landscape (land cover, patch size), its ecological value (diversity, age of certain land use systems) and agricultural management practices. The structural, ecological and management quality of the landscape in Flanders was assessed by multiple indicators (Table 1). The geodatabases and remote sensing data for calculating these indicators are the Flemish land cover map, the so-called *boswijzer*, providing information on forests, the protected nature area geodatabase, the land parcel identification system, the biological valuation map, the small landscape elements geodatabase, the agri-environmental agreements geodatabase and Landsat NDVI data. The small landscape elements geodatabase an the agri-environmental agreements geodatabase are available at the Department of Agriculture and Fisheries of Flanders and the Flemish Land Agency upon request, respectively. Landsat NDVI data were accessed via google earth engine. All other databases are publicly available on the central gateway to geographic government information of Flanders, geopunt.be. A detailed description of the used databases can be found in the Supplementary Materials. Some indicators were available for multiple years, for these indicators the evolution over time was evaluated (see data availability, Table 1).

### 2.1.2. Integration of Landscape Attractiveness Indicators

The landscape attractiveness indicators were integrated to determine an overall landscape attractiveness score for the rural and urban areas of the seven agricultural regions of Flanders (Figure 1). Since no urban area is located in the Weidestreek, the integration of landscape attractiveness indicators was calculated for 13 geographical areas. For each of the 13 geographical areas the considered indicator was ranked from low to high concerning its contribution to landscape attractiveness (see Table 1, landscape attractiveness ranking). Each ranked indicator was scored from 1 to 13, and the geographical area with the highest rank (i.e., the area where the indicator had the most positive impact on landscape attractiveness) received the highest score. The scores of the indicators describing structural, ecological and management quality were averaged resulting in a landscape attractiveness score for structural, ecological and management quality. Finally, these three quality scores were averaged into one overall landscape attractiveness score. Note that the landscape attractiveness scores were calculated for one point in time since some indicators were only available for one year (Table 1). The different aspects of the landscape, hence the studied landscape indicators, were weighted differently by different people [10,33]. For simplicity, it was assumed that each indicator contributed equally to landscape attractiveness. However, indicators that comprised similar information were not all included in the integrated calculation. For example, only the proportion of ≥10 years old grasslands was included since its effect on landscape quality was expected to be higher compared to younger grasslands (i.e., 10–5 G and <5 G). LCcrop was only included for the three most common crop groups: grassland, maize and the crop group grains, seeds and legumes.

**Table 1.** Description of indicators used to characterize the structural, ecological and management landscape quality aspects. Data availability indicates the years where the landscape indicator was available. The data source indicates the geodatabases and remote sensing data used for the calculation of these indicators (see Supplementary Materials). "Included in landscape attractiveness score" indicates if the indicator is used for the calculation of the structural quality, ecological quality and management quality scores and the overall landscape attractiveness score. Year: year of the indicator used for the landscape attractiveness score calculation. Landscape attractiveness ranking: + higher values of the indicator have a positive effect on landscape attractiveness and are higher ranked, − higher values of the indicator have a negative effect on landscape attractiveness and are lower ranked, and references.

| Indicator | Abbreviation | Data Availability | Data Source | Included in Landscape Attractiveness Score | Year | Landscape Attractiveness Ranking |
|---|---|---|---|---|---|---|
| **Structural Quality** | | | | | | |
| Proportion of land covered by protected nature areas | LCnat | 2008-2018 | Protected nature areas geodatabase | yes | 2012 | + [10,24,25] |
| Proportion of land covered by forest | LCfor | 2012 | Boswijzer | yes | 2012 | + [10,24,25] |
| Proportion of land covered by agriculture | LCagri | 2008–2018 | LPIS | yes | 2012 | + [10,24,25] |
| Proportion of agricultural land covered by specific crop groups | LCcrop | 2008–2018 | LPIS | only crop groups grassland, maize, and the crop group grains, seeds and legumes are included | 2012 | grasslands +, maize −, grains, seeds and legumes + [10,26] |
| Proportion of land covered by human infrastructure | LCinf | 2012 | Flemish land cover map | yes | 2012 | − [10,27] |
| Land cover edge density | LCED | 2012 | Flemish land cover map | yes | 2012 | − [10] |
| Land cover diversity (Shannon diversity index) | LCD | 2012 | Flemish land cover map | yes | 2012 | + [5,7,28] |
| Ecological Quality | | | | | | |
| *Biological value of landscape elements* | | | | | | |
| Proportion of biologically very valuable land | BioVV | 2018 | Biological valuation map | yes | 2018 | + [7] |
| *Crop diversity indices* | | | | | | |
| Crop diversity Shannon diversity index | CDSD | 2008–2018 | LPIS | yes | 2018 | + [7,29] |
| Crop diversity Shannon equitability index | CDEI | 2008–2018 | LPIS | yes | 2018 | + [7,29] |
| *Age of grasslands* | | | | | | |
| Proportion of ≥10 years old grasslands | 10G | 2008–2018 | LPIS | yes | 2018 | + [26] |
| Proportion of 10–5 years old grasslands | 10-5G | 2008–2018 | LPIS | no | | |
| Proportion of <5 years old grasslands | <5G | 2008–2018 | LPIS | no | | |
| **Management Quality** | | | | | | |
| Proportion of agricultural fields having small landscape elements | SLE | 2015 | Small landscape elements geodatabase | yes | 2015 | + [5,30,31] |
| Number of agri-environmental agreements per 1000 ha of agricultural land in the period 2008–2020 | AEA | 2008–2020 | Agri-environmental agreements geodatabase | yes | 2018 | + [10,31,32] |
| Fraction of bare fields in winter | fBFw | 2011–2018 | Landsat-7 and -8 + LPIS | yes | 2018 | − [5,32] |

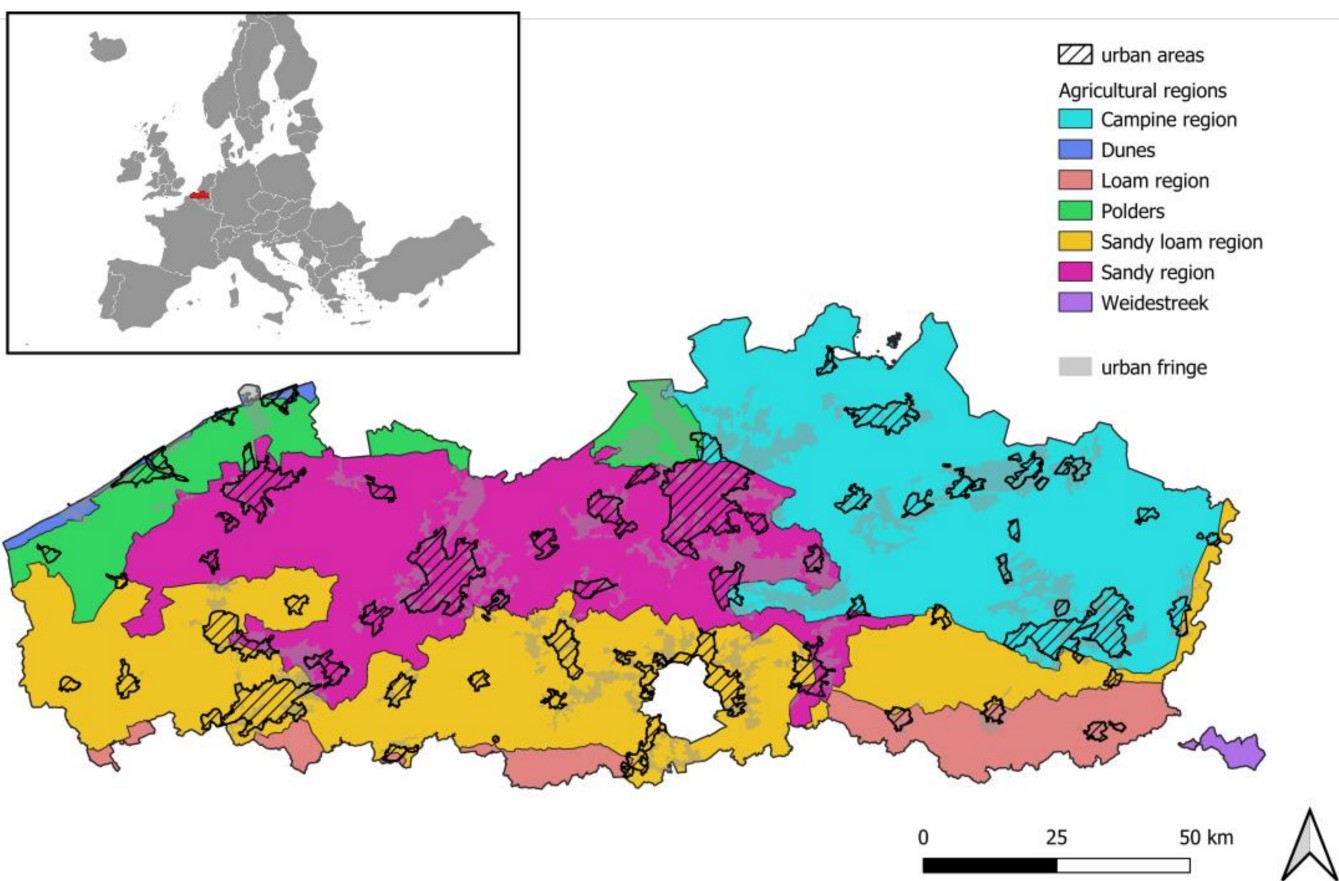

**Figure 1.** Map of the seven agricultural regions, urban (shaded area) and rural (non-shaded area) areas in Flanders. The urban fringe (as defined by [22]) is visualized in grey for reference. Flanders is highlighted in red on the reference map of Europe in the left upper corner.

The interpretation of the indicators with respect to landscape attractiveness (i.e., positive or negative, see Table 1, landscape attractiveness ranking) was based on existing research of relations between landscape attractiveness and landscape characteristics. For instance, several studies have shown that agricultural landscapes with a high diversity in crops, especially diversity in their physical appearance, are found to be more attractive [5,28] and more resilient [29]. Three-dimensional landscape elements such as tree lines, hedgerows and other so-called small landscape elements add variety and character to monotonous agricultural landscapes making them more aesthetically pleasing [5,30], and they provide habitat opportunities for a diverse range of animal species. The authors of [26] demonstrated that the price tourists are willing to pay is positively influenced by the presence of grasslands and negatively by intensive maize cultivation, as the latter obstructs the view. In addition, grasslands are an important sink for atmospheric carbon and support biodiversity, both functions increasing with their age. The naturalness of the landscape, i.e., landscapes with natural elements such as forests, agriculture and protected nature areas, is known to impact landscape aesthetics positively [10,24,25]. Constructions, such as motorways, infrastructure and urban elements are often seen as landscape disturbing elements [10,27]. The implementation of AEA, presence of SLE and the presence of cover crops during winter are examples of practices that can be interpreted as the presence of order and care in the landscape (also referred to as stewardship), which has been shown to impact landscape aesthetics positively [10,31,32]. Edge density is a measure of landscape complexity and landscape disturbance, which is negatively linked with landscape aesthet-

ics [10]. The interpretation of each indicator in terms of landscape attractiveness, and thus its effect on the integrated landscape attractiveness score, is summarized in Table 1.

### 2.2. Scales of Analysis

Flanders is a densely populated, highly fragmented and heterogeneous region. Identifying trends and patterns in landscape attractiveness for such a dynamic and heterogeneous area is challenging. Therefore, as a first step, the region was stratified into sub-regions expected to be characterized by a particular landscape. Two dimensions were considered here during this stratification, i.e., urbanization and the geophysical environment.

Flanders is a region highly affected by urbanization, which is a major driver of the landscape's visual appearance and attractiveness. Future trends in landscape attractiveness can be revealed by examining past trends in the urban and rural landscape of Flanders. The delineation of urban and rural areas as defined in the spatial structure plan of Flanders of 2019 was used (Figure 1). Different spatial policies are in place for urban and rural areas in Flanders, resulting in distinct differences in landscape appearance. Though Flanders is often regarded as one big city, large differences in land cover composition exist between urban and rural areas (Table 2).

**Table 2.** Composition of Flanders regarding rural area, urban fringes and urban areas and their characteristics in terms of land cover (data applies to 2013 and was extracted from [22]). This division of Flanders in rural, urban fringe and urban areas does not aim to be directly applicable for spatial policy [22].

|  | **Rural** | **Urban Fringe** | **Urban** |  |
|---|---|---|---|---|
| % area of Flanders | 79.7 | 13.0 | 7.3 |  |
| % inhabitants | 39.1 | 20.0 | 40.9 |  |
| % buildings | 48.4 | 21.7 | 29.9 |  |
|  | **Rural** | **Urban Fringe** | **Urban** | **Flanders** |
| % settlement area | 23 | 61 | 85 | 32.5 |
| % cropland | 36.0 | 10.1 | 3.3 | 30.2 |
| % grassland | 24.8 | 13.8 | 5.3 | 21.9 |
| % forest | 11.4 | 7.8 | 2.7 | 10.3 |
| % other | 4.8 | 7.3 | 3.7 | 5.1 |

Aside from urbanization, the geophysical environment strongly determines the visual appearance of the landscape in Flanders. The term geophysical environment is used to indicate the combination of soil properties, geological characteristics, physical landscape factors (e.g., relief) and climate conditions. Together, these factors determine the types of land use and the agricultural activities taking place, and consequently the landscape. Seven distinctive agricultural zones are identified in Flanders, i.e., the Campine region, Dunes, Loam region, Polders, Sandy Loam region, Sandy region and Weidestreek (Figure 1). The Campine and Sandy regions are characterized by sandy soils. The Campine region has a more continental climate compared to the neighboring regions. In the Sandy Loam region, sandy and sandy loam soils prevail. In the Loam region, loam soils are most common and the topography is undulating and characterized by small hills. The Weidestreek is known for its pasture landscape, where loamy and stony soils are common. In the Polders and the Dunes, the coastal regions of Flanders, more heavy soil textures such as clay prevail. The Polders and the Dunes have a more maritime climate compared to the other agricultural regions. The regions were defined in 1951 and modified in 1953, 1968 and 1974. The agricultural regions are considered in several agricultural policies, for example, for the determination of the maximum lease price of agricultural land. As a result, landscape appearance is expected to be similar within each of the regions and considerably different between the regions.

The resulting stratification, as shown in Figure 1, allowed us to evaluate landscape attractiveness spatially and to elucidate the most important trends of regional landscape attractiveness.

## 3. Results

### 3.1. Structural Quality Indicators

#### 3.1.1. Land Cover Structural Quality Indicators

Agriculture dominated the landscape of Flanders in 2012, whereas forest and nature were much less prominent (Table 3). The share of agricultural land was highest for the Loam region, Weidestreek and Polders. As a result, these regions were characterized by a relatively low degree of fragmentation (depicted by the edge density, LCED) and were less diverse compared to the other regions. Agriculture was less prominent in the Campine region and the Dunes in 2012. The Campine region stood out for its high share of forests and hence, obtained the highest land cover diversity. The Dunes region was the area most heavily affected by urbanization, followed by the Sandy region, whereas the Weidestreek clearly had the least proportion of impervious surfaces and more nature compared to the other regions.

**Table 3.** Structural quality indicators (data applies to 2012 and was extracted from [22]), ecological quality indicators and management quality indicators for Flanders, urban and rural areas, and the seven agricultural regions. (see Table 1 for description of indicators).

| | Flanders | Urban | Rural | Dunes | Campine Region | Loam Region | Polders | Weidestreek | Sandy Loam Region | Sandy Region |
|---|---|---|---|---|---|---|---|---|---|---|
| **Structural Quality** | | | | | | | | | | |
| LCnat | 1.00 | 0.57 | 1.05 | 0.99 | 1.87 | 0.80 | 1.15 | 2.48 | 0.51 | 0.63 |
| LCfor | 12.38 | 8.47 | 12.85 | 5.47 | 25.86 | 4.27 | 0.83 | 18.35 | 7.29 | 8.89 |
| LCagri | 51.60 | 13.27 | 56.25 | 21.90 | 37.90 | 71.57 | 66.13 | 66.55 | 58.35 | 50.55 |
| LCinf | 16.21 | 43.5 | 12.94 | 31.69 | 15.67 | 10.80 | 14.71 | 6.07 | 15.44 | 19.07 |
| LCED | 0.01 | 0.04 | 0.03 | 0.03 | 0.03 | 0.03 | 0.02 | 0.02 | 0.03 | 0.03 |
| LCD | 1.03 | 1.26 | 1.32 | 1.41 | 1.46 | 1.19 | 1.35 | 1.20 | 1.35 | 1.44 |
| **Ecological Quality** | | | | | | | | | | |
| BioVV | 5.3 | 2.7 | 5.7 | 27.7 | 8.0 | 2.9 | 2.2 | 13.4 | 4.9 | 3.9 |
| **Management Quality** | | | | | | | | | | |
| SLE | 7.6 | 13.7 | 7.5 | 5.0 | 12.3 | 5.6 | 3.1 | 9.5 | 6.2 | 7.8 |
| AEAs | 5.9 | 5.5 | 5.9 | 2.2 | 4.1 | 12.3 | 8.7 | 29.7 | 6.7 | 2.8 |

#### 3.1.2. Nature and Forest Structural Quality Indicators

Over the course of ten years, the total area of protected nature areas increased by 5582 ha, representing 1.33% of the total area of Flanders in 2018. The proportion of land covered by protected nature areas (LCnat) was highest in the Weidestreek, followed by the Campine region and, to a lesser extent, the Polders (Table 3). Relatively few nature areas were located in the Sandy Loam region. Except for the Weidestreek and Dunes regions, where LCnat remained stable, nature reserves steadily expanded in all regions, albeit at different rates (Figure S1). Increases in LCnat were most prominent in the Campine region and Sandy Loam region, and modest in the Loam region (Figure S1).

The forested area in Flanders slightly increased between 2009 and 2012 by 0.32%, only to decrease again in the following three years by 0.41% (Figure S1).

### 3.1.3. Agriculture Structural Quality Indicators

The proportion of land covered by agriculture (LCagri) in Flanders was 50.9% in 2018. This high percentage indicated the important role of agriculture in the landscape of Flanders. Grassland, maize and the crop group grains, seeds and legumes were the three crop groups with the highest crop specific proportion of agricultural land covered (LCcrop) in 2018, representing 35.0%, 26.2% and 12.0% of the agricultural area of Flanders in 2018, respectively. The area of land covered by agriculture in Flanders decreased by 10,201.39 ha between 2008 and 2018. LCagri decreased by 1.5% in ten years. The change in LCcrop between 2008 and 2018 varied for the different crop groups (Figure S2). The LCcrop of the crop groups: maize, grains, seeds and legumes, grassland and other crops decreased the most in the last years (Figure S2). The largest increases were found for the crop groups: potatoes, vegetables, herbs and ornamental plants, fodder crops and the crop group fruit and nuts (Figure S2).

LCagri was much lower in urban areas than in rural areas in 2018. Only 11.9% of the urban area was occupied by agriculture, whereas this was 55.7% in rural areas. Field sizes were smaller in urban areas (0.679 ha) than in rural areas (0.850 ha) in 2018. The three crop groups with the highest LCcrop were the same in both urban and rural areas of Flanders 2018: grassland, maize and the crop group grains, seeds and legumes (Figure 2). The fraction of agricultural area occupied by grassland was higher in urban areas (47.6%) than in rural areas (34.7%) in 2018, with the opposite for maize and the crop group grains, seeds and legumes (maize: 23.8% in urban and 26.3% in rural areas; grains, seeds and legumes: 7.8% in urban and 12.1% in rural areas).

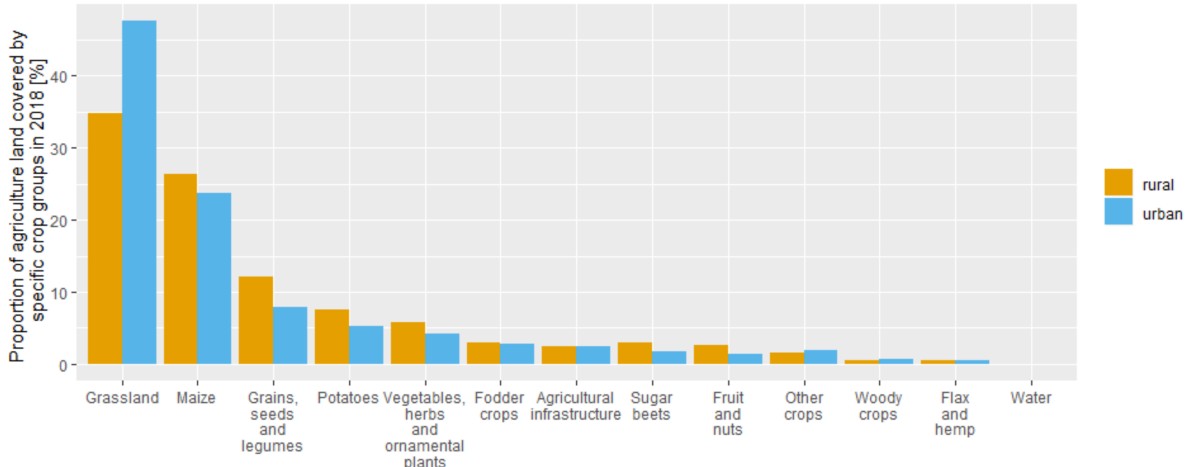

**Figure 2.** Proportion of urban (blue) and rural (orange) agricultural land covered by specific crop groups (LCcrop) in 2018. (Data source: LPIS).

The change in agricultural area from 2008 to 2018 was clearly different between rural and urban areas (Figure 3). In urban areas the agricultural area decreased strongly in the period from 2008–2018, whereas this was not the case in rural areas.

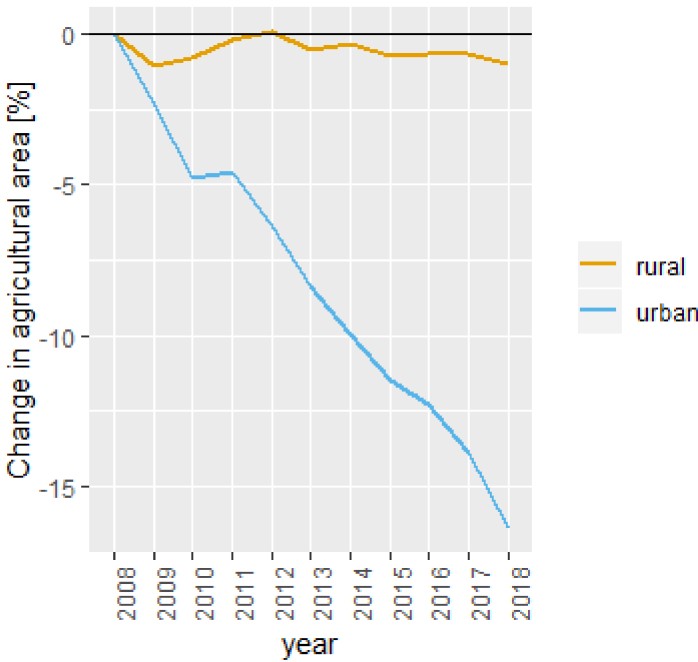

**Figure 3.** Change in agricultural area from 2008 to 2018 in rural (orange) and urban areas (blue). The change was calculated as = ((agricultural area in year i-agricultural area in 2008)/agricultural area in 2008) × 100. (Data source: LPIS).

In all agricultural regions, except in the Loam region, grassland was the most dominant crop group in 2018 (Figure S3). The proportion of agricultural land covered by grassland in 2018 was the highest in the Weidestreek (61.5%) and the lowest in the Sandy region (30.6%). In the Loam region, the crop group grains, seeds and legumes had the highest LCcrop.

*3.2. Ecological Quality Indicators*

3.2.1. Biological Value of Landscape Elements

The proportion of biologically very valuable land (BioVV) was equal to 5.3% in Flanders (Table 3). Only 2.7% of the urban land cover was classified as BioVV. In rural areas, BioVV was equal to 5.7% (Table 3).

BioVV varied between the different agricultural regions. BioVV was highest in the Dunes (27.7%) and lowest in the Polders (2.2%) (Table 3).

3.2.2. Crop Diversity Indices

The crop Shannon diversity index (CDSD) only slightly decreased in the Dunes, Campine region, Weidestreek and Sandy region, and slightly increased in the Loam region and Polders between 2008 and 2018 (Figure 4a). The Loam region, which is mostly dominated by agriculture (cf. Table 3), was also the region with the highest CDSD, shortly followed by the Sandy Loam region. The Weidestreek clearly had the lowest CDSD, being strongly dominated by grassland (around 61% of its total agricultural area). The crop diversity Shannon equitability index (CDEI) (Figure 4b) falls within the range of 0.4–0.6 for all regions, indicating that crop types were relatively unevenly distributed within the regions (an index value of 1 indicates perfect evenness). An even more important observation is that CDEI clearly declined between 2008 and 2018 in all regions (Figure 4b).

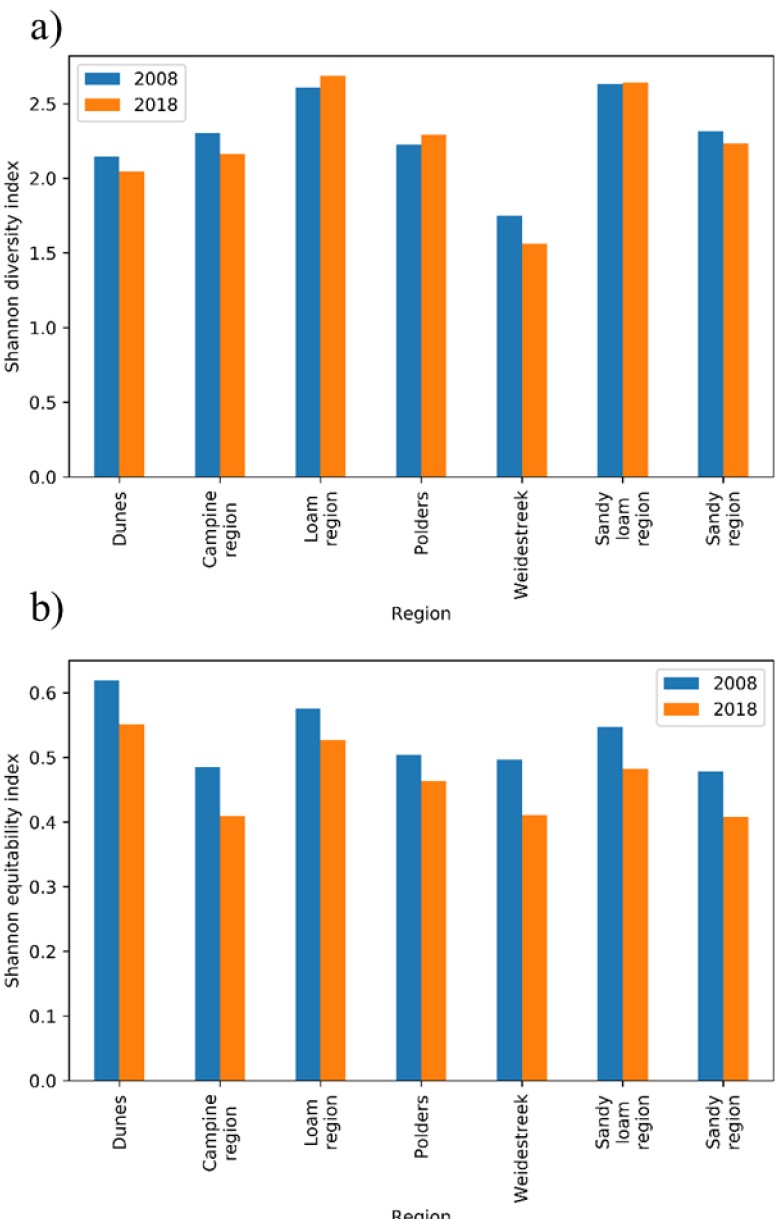

**Figure 4.** Comparison of crop diversity between the different agricultural regions in Flanders between 2008 and 2018, as measured by (**a**) the crop Shannon diversity index (CDSD) and (**b**) the crop diversity Shannon equitability index (CDEI). (Data source: LPIS).

### 3.2.3. Age of Grasslands

In 2018, roughly half (52.7%) of all grassland fields in Flanders fell into the category of ≥10 years old grassland, i.e., being under grassland management for at least 10 consecutive years. Grassland aged less than 5 years was the second largest category, accounting for 32.2% of all grassland fields. The remaining 15.1% was made up of 5 to 10 years old grassland. This distribution turned out to be stable throughout Flanders. No significant differences were found between urban and rural areas in this respect. Regarding the agricultural regions, only the Campine region and Weidestreek deviated from this overall trend. In the Campine region, only 40% of the grassland fields could be categorized as ≥10 years old grasslands, whereas in the Weidestreek, ≥10 years old grasslands were far more common practice and amounted to 64.3% of all grasslands.

*3.3. Management Quality Indicators*

3.3.1. Small Landscape Elements

The proportion of agricultural fields having small landscape elements (SLE) in Flanders was equal to 7.6% in 2015 (Table 3). SLE in urban agricultural fields (13.7%) were almost double compared to rural fields (7.5%). SLE also varied between the different agricultural regions (Table 3). SLE were highest in the Campine region, 12.3% of its agricultural field area was occupied by small landscape elements, and lowest in the Polders (Table 3).

3.3.2. Agri-Environmental Agreements

The number of agri-environmental agreements in Flanders (AEA) increased from 2140 to 7816 between 2008 and 2020. Between 2008 and 2011, AEA steadily increased from roughly 2000 in 2008 to 4000 in 2011. Between 2011 and 2019, AEA remained stable. Compared to 2019, a strong increase in the number of AEA was observed in 2020. Roughly the same trends occurred in urban versus rural areas, the former only containing a small fraction of the total amount of measures implemented in Flanders. The abundance of AEAs relative to the area of agricultural land showed clear differences between the agricultural regions (Table 3). The Weidestreek especially stands out with a high abundance of AEA compared to the other regions, followed by the Loam region and Polders. These regional differences in AEA and management practices could explain differences in visual appearance between the different regions.

3.3.3. Fraction of Bare Fields in Winter

The fraction of bare fields during wintertime decreased over time, irrespective of the crop group (Figure 5). We consider this observed decreasing trend in bare fields to have a positive impact on landscape attractiveness. One large exception in this overall trend was the winter of 2016. Climatologically, this winter was very dry, explaining the fierce drop in average NDVI values across agricultural fields and leading to high quantities of detected bare fields.

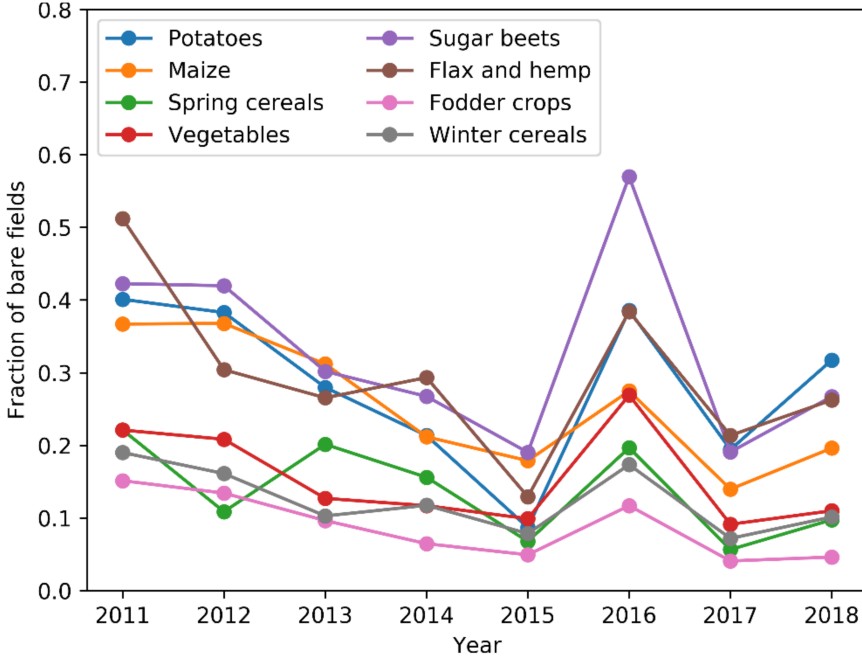

**Figure 5.** Evolution of fraction of bare fields in winter (November–January) attributed to the main summer cultivation for all major crop groups in Flanders. Note that the dry 2016 winter resulted in very low NDVI values and a potential under-detection of covered fields. (Data source: Landsat NDVI).

The use of cover crops was found to be dependent on the crop type grown on the field before the winter. Parcels used for growing vegetables, fodder crops, winter cereals and spring cereals tended to be more frequently covered by a cover crop in winter compared to the other major crop groups.

Overall, the share of winter cover crops did not differ between urban and rural areas but differed between the agricultural regions. The northern Campine and Sandy regions had the lowest share of bare fields in winter (mostly below 20%), followed by the southern Loam and Sandy Loam regions, which had an intermediate share of bare fields of mostly below 40%. The western regions near the coast (Polders and Dunes) had the highest share of bare fields (around 50%). The differences between the regions are largely related to the regulations for the latest sowing and earliest ploughing of cover crops, with a minimal interval: Polders and Dunes [20/8, 15/10], Loam [1/10, 30/11], and Sandy Loam and other regions [30/10, 31/1]. During these periods catch crops or green cover is maintained on the fields.

### 3.4. Integration of Landscape Attractiveness Indicators

The integrated landscape attractiveness scores are visualized in Figure 6. In urban areas, the structural quality and ecological quality scores were lower compared to rural areas. The opposite was true for the integrated management quality score. The latter was mainly caused by the higher number of small landscape elements in urban areas. The largest contrast in landscape attractiveness scores between urban and rural areas was found in the Dunes and the Sandy region for the structural quality score and the ecological quality score, respectively. In general, the contrast between urban and rural scores from a management point of view was lower compared to the differences in structural and ecological quality scores. Landscape attractiveness scores varied between the different agricultural regions (Figure 6). The rural areas of the Campine region and the Weidestreek were the regions with the highest overall score on landscape attractiveness, whereas in the rural areas of the Dunes, Polders and the Sandy region, landscape attractiveness was rather low.

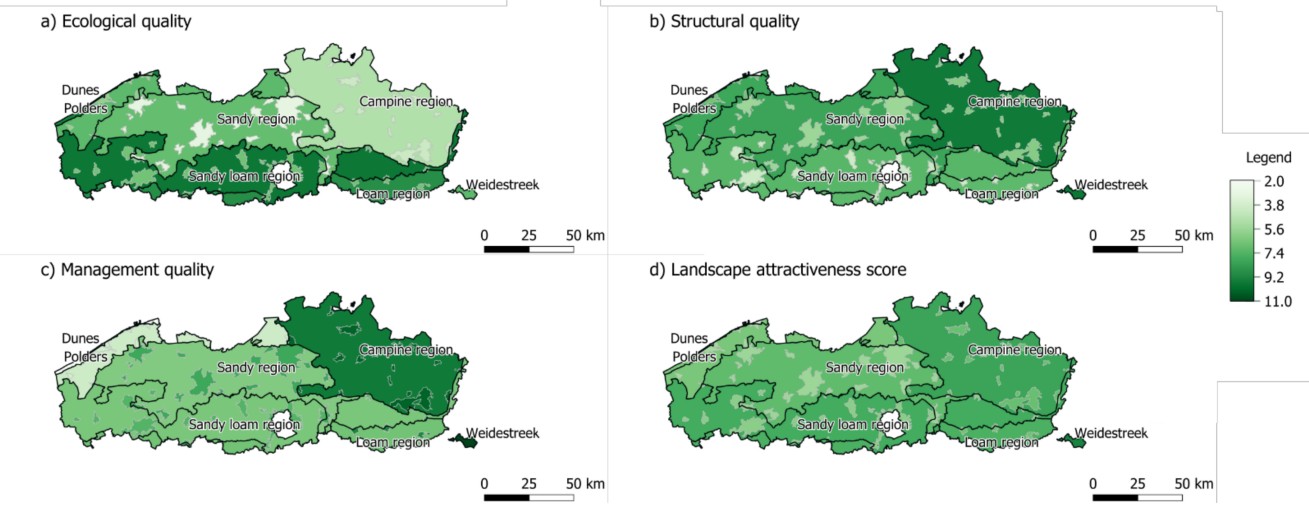

**Figure 6.** (**a**) Ecological quality score map, (**b**) Structural quality score map, (**c**) Management quality score map and (**d**) overall landscape attractiveness score map. Indices were calculated for the urban and rural area in each agricultural region. Agricultural regions were delineated with a black line, urban regions with a grey line. Darker green regions have a higher scoring on the considered landscape attractiveness score.

## 4. Discussion

Landscape attractiveness was assessed using a spatial approach in (i) Flanders, (ii) rural versus urban areas and (iii) the seven agricultural regions of Flanders, and

discussed in the landscape policy framework in the following sections. Temporal changes in certain aspects of landscape attractiveness were elucidated from the landscape indicators that were available for multiple years.

### 4.1. Landscape Attractiveness in Flanders

In Flanders, where agriculture dominates the landscape, the area occupied by the agricultural sector decreased from 50.9% to 49.4% between 2008 and 2018. The number of agricultural holdings in recent years also decreased in Belgium, amounting to 68% between 1980 and 2018 [34,35]. The area that was previously used by agriculture is now mostly used by other economic sectors [36]. In rural areas in Flanders, 64% of the established enterprises are not related to agriculture [37]. The impact on the landscape of this trend of increasing numbers of non-agricultural enterprises in rural areas varies depending on the established enterprises. In some cases, non-agricultural enterprises in rural areas might have far-reaching negative effects on landscape attractiveness. For example, when newly established enterprises require large outside storage areas, this results in more traffic or noise nuisance [36]. In cases where the newly established enterprises' activities take place in already existing buildings, the effect on the landscape is minimal. In addition, for some non-agriculture enterprises the rural character and landscape attractiveness are important, for example, for bed and breakfasts, yoga studios or conference halls that choose to establish themselves in rural areas because of the rural landscape itself [22]. Another element that can explain the decrease in area under agriculture is the economic diversification of agricultural businesses themselves. More and more agricultural businesses are trying to diversify their income by establishing, for example, farm shops, care farms or rural tourism.

In total, 73.3% of the agricultural land in Flanders consisted of grasslands, maize, grain, seed and legume fields in 2018. The dominance of these crop groups in Flanders is not surprising as all these crop groups can be linked with livestock farming, which is an important agricultural sector in Flanders [38]. The high share of these three crop groups moreover indicates that the agricultural landscape diversity in Flanders is rather low. This is confirmed by the low crop diversity Shannon equitability index (CDEI). In addition, the observed decrease in CDEI between 2008 and 2018 indicates that agriculture is becoming even more and more focused on a handful of crops in Flanders. Generally, less biodiverse landscapes are perceived as being less attractive [39]. In the current agricultural policy framework of Flanders, policies dealing with biodiversity loss and climate change are included. In addition, with the recent launch of the European Green Deal, an even stronger focus on agricultural policies dealing with biodiversity loss and climate change is expected. These policies will likely influence the agricultural landscape in Flanders.

Some shifts in the agricultural landscape of Flanders are related to grasslands. This is important for the landscape attractiveness of Flanders as grassland dominates the agricultural landscape (i.e., 35% of the agricultural land was grassland in 2018). First, a clear decrease in the median field size of grasslands in Flanders was observed. The number of grassland fields bigger than 1.5 ha decreased from 53,093 in 2008 to 48,858 in 2018. The decrease in grassland area in Flanders is thus mostly caused by the disappearance of large grassland fields. Grasslands are becoming less and less prominent in the landscape of Flanders. These observed dynamics in grasslands also underpin the need of the current LULUC agricultural policies in Flanders that aim to increase soil carbon storage in order to increase the climate resilience of the landscape. For example, current policies aim to stimulate land use that increases soil carbon storage such as the conversion of arable land to extensive grassland.

From a landscape management perspective, we have noticed positive trends in Flanders over the past ten years. Overall, the number of AEA established and the fraction of green covered fields in winter increased enormously. Both trends contribute to the presence of order and care in the landscape (also referred to as stewardship) and, as such, have a positive impact on its attractiveness [10]. These trends can be explained by recent efforts to focus on a participatory and holistic approach in landscape planning and management.

Farmers are no longer perceived as mere food suppliers but are increasingly recognized as key stakeholders in the landscape and more frequently involved in landscape planning, conservation and rehabilitation projects. Policy instruments such as the AEA are key drivers of this participatory approach and clearly have had a positive impact on the landscape over the past years.

### 4.2. Landscape Attractiveness in Urban Versus Rural Areas

From a general land cover perspective, urban and rural areas obviously differ substantially from one another (Table 2). Therefore, in the remainder of this section, we focus on comparing specific landscape indicators between these two extremes in Flanders. This might provide us some clues as to how the landscape will be transforming in the future given the continuing urbanization trend.

Several observations indicate an increased conversion from agricultural area to urban fabric in the urban fringes, demonstrating that urban areas are losing agricultural landscape elements faster than rural areas. First, the decrease in agricultural area was found to be stronger in urban areas compared to rural areas. This makes sense, as the pressure for (open) space is higher in urban areas than in rural areas. In addition, the use of former agricultural land for hobby farming and horse keeping is found to be higher in and around urban areas than in rural areas in Flanders [22,40]. Second, the median field size and the area of agricultural fields was lower in urban areas compared to rural areas. This demonstrates that agriculture is less prominent in the urban landscape and more fragmented than in the rural landscape. The urban population is attracted to the stronger agricultural character of the landscape in rural areas compared to urban areas. Rural landscape consumption by the urban population takes different forms, such as recreation, leisure and outdoor activities or interest in the establishment of a second residence [6]. This variety of consumption demands by urban population influences rural landscape dynamics [6,41] and will consequently affect rural landscape attractiveness.

When we look at the biological value of the landscape, we see a difference between urban and rural areas. The area of biologically very valuable landscape is low in rural areas, only amounting to 5.7% of the total rural area. In urban areas this fraction is even lower (i.e., 2.7%), indicating a smaller natural habitat area in urban areas. Several studies have already demonstrated that habitat loss has a negative effect on biodiversity [42]. In addition, the higher landscape fragmentation, as confirmed by the higher LCED in urban areas compared to rural areas (Table 3), suggests that natural habitats are also more fragmented (i.e., broken apart [42]) in urban compared to rural areas. This has a direct impact on landscape attractiveness, as green spaces with high biological diversity, which are more likely to occur in biologically very valuable landscapes, are valued higher than green spaces with lower biological diversity [7]. In addition, both crop diversity and the fraction of grasslands > 10 years old were found to be consequently lower in urban areas compared to rural areas. All these observations related to a low biodiversity in urban compared to rural areas, once again underline the need for strong agricultural policies, especially in the urban context, with a strong focus on conserving and stimulating biological diversity.

A higher presence of small landscape elements in agricultural fields in urban areas compared to rural areas was found. Because of the low presence of green elements in the urban landscape, the disappearance of agricultural fields with small landscape elements in urban areas is likely to result in protest or petition from the population in these areas more easily than in rural areas. This can explain the higher presence of small landscape elements in urban agricultural fields. Furthermore, AEA are equally well adopted and established in urban versus rural areas, which is an encouraging finding, meaning that also in highly urbanized regions, agricultural activities are being conducted with the necessary care for the immediate environment. In an increasingly urbanizing context, safeguarding the quality of the remaining agricultural land will indeed be an important priority for future urban policy.

*4.3. Differences in Landscape Attractiveness between Agricultural Regions*

When we look at the structural, ecological and management scores (Figure 6) we see that regions scoring low for one landscape quality aspect do not necessarily score low for the others as well. Each region is characterized by a distinctive geophysical environment, resulting in different land cover compositions (Table 3) and land management practices. The latter is for instance reflected by different crop distributions across regions and different adoption patterns of AEA. As a result, each region has its own specific merits and challenges, highlighting the need for a spatially differentiated policy approach. Low scoring on a specific landscape quality aspect can indicate potential risks and point to the need for specific policy measures. Regions scoring low on the structural landscape quality aspect (predominantly urban regions) are characterized by a high degree of soil sealing and low land cover diversity. As a result, these regions are faced with a low water buffering capacity and should, considering climate change, focus on sustainable water management, enabling them to better cope with both excess and scarcity of water in the future. Regions characterized by a low ecological quality score (e.g., the Campine region) should prioritize sustainable grassland management, enhance crop diversity and encourage the adoption of AEA by farmers. Regarding the latter, considerable progress has been made in the past year (see Section 3.3.2). Finally, the coastal regions, scoring low on management quality, mostly require an effort regarding implementation of small landscape elements, cover crops in winter and AEA.

In general, we see a growing consciousness amongst farmers regarding their environmental impact, leading to an increased engagement in AEA and adoption of winter cover crops. This is an encouraging trend of the past few years, given the ambitious goals of European rural legislations and the European Green Deal. However, considerable regional differences in the presence of small landscape elements and AEA do remain. The occurrence of these management practices is nowadays still strongly linked to regional geophysical characteristics and traditions. For instance, hedgerows are mostly planted in the Weidestreek, where they have long been part of the landscape as part of a traditional and historical land management system. However, given their clear benefits with respect to the resilience, biodiversity and visual appeal of the landscape, other regions in Flanders should not hesitate to adopt this practice as well. Cover crops, however, provide many more benefits (increased biodiversity and soil carbon storage) and should therefore become a more integral part of agricultural management practices, irrespective of the location. In conclusion, there is a growing need to diversify agricultural management practices in the different regions. Studies looking into the incentives for farmers to engage in and adopt such management practices (e.g., [43]) are therefore crucial for policymakers, enabling them to define support programs for farmers with a maximum chance of success.

## 5. Conclusions

The current state of landscape attractiveness in Flanders was assessed using an objective monitoring framework for landscape attractiveness based on a set of indicators derived from geospatial databases and remote sensing data. We were able to identify trends related to agriculture in Flanders, such as a slight decrease in total agricultural area, a decrease in the dominance of grassland, maize and cereals, a decrease in crop diversity, an enormous increase in the adoption of AEA and a decrease in bare soil conditions during the winter. In particular, the latter two trends point to an increasing environmental concern of farmers and are key in producing climate-resilient and biodiverse agricultural landscapes. With the ever-increasing availability of geospatial data on various land-related topics (e.g., land cover, biological valuation) these analyses will become increasingly easier to repeat in the future, enabling a dynamic and holistic assessment of landscape quality trends over time, considering all of its aspects. If the geodatabases used to derive landscape attractiveness indicators were available on a yearly (or 5 yearly) scale, an operational service to evaluate landscape attractiveness could be developed. This would also allow for the monitoring of

temporal changes in specific landscape indicators, which was currently not possible for all indicators (see Table 1 on data availability).

Spatial differences in landscape quality across Flanders were observed. First of all, urban areas clearly showed a lower quality overall (with management quality being the sole exception due to a higher presence of small landscape elements). In the context of a strongly urbanized region such as Flanders, where urbanization has been severely spreading into rural areas, this calls for dedicated policy actions to prevent further loss of landscape quality in these urban fringes. Secondly, our analyses have demonstrated large local differences in landscape quality between the seven agricultural regions in Flanders. The calculated landscape attractiveness scores provide more insights into the merits and weaknesses of each region, in turn enabling the definition of policy and management priorities to boost landscape attractiveness in each region separately. This clearly demonstrates that a localized approach, considering the geophysical reality of each individual region, is required to improve landscape quality in Flanders as a whole.

To conclude, the observed trends of multi-annual indicators and spatial variation in landscape attractiveness in Flanders, in combination with a dedicated policy analysis, can be used as a framework to assess the potential effects of future policy plans, identify policy gaps and evaluate past landscape policy. More specifically, the framework can be used in a SWOT analysis, which is often performed in the situation analysis stage of the policy developing process, as a conversation starter in discussions with stakeholders, to identify regions where specific policy development is (urgently) needed, and for the evaluation of (past) policy programs.

**Supplementary Materials:** The following are available online at https://www.mdpi.com/article/10.3390/land10070703/s1. The data sources used for the calculation of indicators (see Table 1) are discussed in the Supplementary Materials file. Refs [44–49] are cited in Supplementary Materials file.

**Author Contributions:** Conceptualization and methodology, A.G., J.D. and A.V.; validation, formal analysis, writing—original draft preparation, A.V. and J.D.; data curation and writing—review and editing, A.G., J.D. and A.V.; supervision, A.G., funding acquisition, A.G. All authors have read and agreed to the published version of the manuscript.

**Funding:** The authors acknowledge funding from the European Union's Horizon 2020 Research and Innovation Programme under grant agreement No. 818496.

**Data Availability Statement:** Publicly available datasets were analyzed in this study. This data can be found here: https://www.geopunt.be/catalogus, accessed on 18 April 2021.

**Acknowledgments:** We thank the editor and the anonymous reviewers for their feedback which helped us to improve the manuscript.

**Conflicts of Interest:** The authors declare no conflict of interest.

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
