# Peer review of "Evaluating Landscape Attractiveness with Geospatial Data, A Case Study in Flanders, Belgium"

_land, doi:10.3390/land10070703_

Round 1

Reviewer 1 Report

REVISION

This is a very interesting manuscript about a topic of environmental conservation concern. Manuscript is well written with a large number of references (but I suggest some others). Data are original and useful for policy actions at landscape level. I have only two major comments. First, I am an applied/operational planners and managers in applied ecology and when I read these papers, I would like to read some operational suggestions following the results obtained. In this regard, I would read less general statements but useful ideas and solutions. Therefore, I suggest to improve the conclusions. Second, statistic is only descriptive and should be improved, at least for some data. Can you add some stat tests of comparisons or correlations (also simple and non parametric?).

Therefore, I suggest MINOR REVISIONS. I would like to read a second revision improved. Here below some minor comments and suggestions that, I hope, could improve a bit, the first draft of the manuscript.

  1. 212. Results are merely descriptive (but very interesting). The authors should add some statistical approaches to test for significance of differences (e.g. ‘representing 35.0%, 26.2% and 12.0% of the agricultural area’: are these differences significant among them?).
  2. 431-432. ‘increased fragmentation and associated pressures caused by human presence and interference’. The authors should better explain the concept of fragmentation, degradation, isolation (see Fahrig, 1997 and others).

The last sentence: ‘To conclude, the observed trends and spatial variation in landscape attractiveness in Flanders in combination with a dedicated policy analysis can be used as a framework to assess the potential effects of future policy plans, identify policy gaps and evaluate past landscape policy.’ is too general. I suggest to specify better how your data can affect project planning and management carried out by Public Agency. For example, in which step of a project cycle (see Hockings et al., 2006, IUCN and Journal for Nature Conservation, 41, 63-72, 2018, for an application in landscape and biodiversity conservation).

In acknowledgments, please add the role of anonymous reviewers and Editor.

Have a nice work.

Author Response

Dear reviewer,

We appreciate the interesting questions and remarks on our manuscript. We revised the manuscript with care and provide a response to your remarks below.

All the best,

The authors

  1. Results are merely descriptive (but very interesting). The authors should add some statistical approaches to test for significance of differences (e.g. ‘representing 35.0%, 26.2% and 12.0% of the agricultural area’: are these differences significant among them?).

The presented data cover the years 2008 - 2018. To be able to perform a multiple comparison statistical test multiple observations per group (i.e. urban versus rural and the seven agricultural regions) are needed. However, for each indicator (Table 1) we only have one observation per group for a particular year. Therefore, no multiple comparison tests were applied. Instead, since our data are spatially inclusive and available for the entire study area, we performed a temporal analysis in terms of anomalies which proved very meaningful to different stakeholders, and in particular policy makers.

431-432. ‘increased fragmentation and associated pressures caused by human presence and interference’. The authors should better explain the concept of fragmentation, degradation, isolation (see Fahrig, 1997 and others).

We agree and have added some clarifying sentences on the concept of fragmentation and its implications; see line 462-466.

The last sentence: ‘To conclude, the observed trends and spatial variation in landscape attractiveness in Flanders in combination with a dedicated policy analysis can be used as a framework to assess the potential effects of future policy plans, identify policy gaps and evaluate past landscape policy.’ is too general. I suggest to specify better how your data can affect project planning and management carried out by Public Agency. For example, in which step of a project cycle (see Hockings et al., 2006, IUCN and Journal for Nature Conservation, 41, 63-72, 2018, for an application in landscape and biodiversity conservation).

We added examples of how and during which stages the observed trends and spatial variation in landscape attractiveness can be used in policy development/evaluation. Lines 554-558

In acknowledgments, please add the role of anonymous reviewers and Editor.

We added this to the acknowledgments, see line 688-689

Reviewer 2 Report

The manuscript entitled “Evaluating landscape attractiveness with geospatial data, a case 2 study in Flanders, Belgium” present the results of a 10 years monitoring of the quality of landscape of Flanders based in a set of selected indicators. The work is scientifically soundness, with technical aspects very detailed. The results support the conclusions, and the general interest is, in my opinion, high, as similar monitoring schemes could be used by administrations in other areas of the world. Although, I am not a native English speaker, I feel the language is correct. I only provide several minor comments:

Line 72. There is a typo in the order. In the text it appears (ii) twice, instead (iii).

Line 96. I think the geospatial and remote sensing data should be briefly mentioned here, with one or two sentences, and maintain as now, a detailed information in Supplementary material.

Lines113-114. Each indicator has the same given value or (+1 or-1). I understand this, so it is necessary to stablish a criterion, but it is strongly possible that some indicators were most important than others. I think this assumption of equality in importance should be briefly mentioned and discussed as possible limitation.

Author Response

Dear reviewer,

We appreciate the interesting questions and remarks on our manuscript. We revised the manuscript with care and provide a response to your remarks below.

All the best,

The authors

Line 72. There is a typo in the order. In the text it appears (ii) twice, instead (iii).

This was changed.

Line 96. I think the geospatial and remote sensing data should be briefly mentioned here, with one or two sentences, and maintain as now, a detailed information in Supplementary material.

We added some sentences on this on lines 98-108.

Lines113-114. Each indicator has the same given value or (+1 or-1). I understand this, so it is necessary to stablish a criterion, but it is strongly possible that some indicators were most important than others. I think this assumption of equality in importance should be briefly mentioned and discussed as possible limitation.

We added more detail on the procedure of the score calculations and addressed your concern on the assumption of equality. See lines 140-162

Reviewer 3 Report

This article tries to evaluate the landscape attractiveness using geospatial data taking the case of Flanders in Belgium. The subject of this research is fascinating and vital from the perspective of landscape design, ecology, and regional planning. This article mainly focused on the quantitative indexing of the landscape features which are related to the attractiveness. However, the attractiveness itself is not defined nor even discussed in depth. Obviously, the “attractiveness” is a subjective concept to evaluate the landscape. The framework to demonstrate why these objective indices can represent or explain the attractiveness is the most crucial point to convince the readers that the proposed approach is helpful for landscape evaluation. In the current style, I feel this article just exhibits the bundle of geospatial features of agricultural or natural land use but does not provide a clear idea on the evaluation of the landscape attractiveness.

I recommend authors discuss their data in conjunction with the existing ideas on evaluating landscape attractiveness, for example, in the field of Environmental Aesthetics or Environmental Ethics.

The following literature may provide the ideas.

Carlson, Allen. 2010. “Contemporary Environmental Aesthetics and the Requirements of Environmentalism.” Environmental Values 19 (3): 289 314.

Brady, Emily. 2011. “The Ugly Truth: Negative Aesthetics and Environment.” Royal Institute of Philosophy Supplement 69: 83 99.

Author Response

Dear reviewer,

We appreciate the interesting questions and remarks on our manuscript. References to relevant research used to interpret of the used landscape indicators concerning their contribution to landscape attractiveness (positive or negative) were added to Table 1 and discussed in lines 140-162.

All the best,

The authors

Reviewer 4 Report

The article is a good case study of an accepted monitoring process 

I have two comments concerning the methods used in the case study.

These are not comments against the case study methods as it just uses available map data and apparently could not undertake any original surveys. They are just comments on the available data about Biological condition that might be mapped in future

  1. The information on soil conditions is really just a physical description, there are no indicators for biodiversity in the soil, soil flora and fauna being excellent indicators of health or condition
  2. Methods of monitoring biological condition and biodiversity by DNA is now common practice, see European Journal of Soil Science Oct 2009. 60, 807-819 Gardi et-al [email protected]

My suggestion is you take this up with the monitoring authorities as being a possible innovation for the future, that is as an observation from your case study

Author Response

Dear reviewer,

We appreciate your remarks on our manuscript. We will communicate your suggestion on the monitoring of soil biodiversity to the monitoring authorities. The revised manuscript can be found in attachment.

All the best,

The authors

Reviewer 5 Report

Contribution

The manuscript provides an analysis and interpretation of landscape attractiveness in Flanders overall, in urban vs. rural regions in Flanders, and in seven different agricultural regions in Flanders. The approach leverages geospatial data and integrates them into landscape attractiveness scores in a novel way for the region.

Significance

  • Broader impact: The results may be used to inform land policies in Flanders.
  • Intellectual merit: Unclear (see Major Concerns below).

Novelty

  • Unclear (see Major Concerns below)

Major Concerns

  • The manuscript in its current form lacks a deep literature review (only 12 citations in the introduction). As a result, it is unclear how the described work builds on or extends previous efforts and how it is theoretically grounded.
  • I am not convinced by the argument to “discard the cultural dimension as it can be regarded as 88 static in a time period of ten years, from 2008 to 2018.” Thinking about politics around the world, including the rise of right-wing populism, quite a bit has changed culturally. I’d like to hear a better argument for discarding the cultural dimension or an inclusion of the cultural dimension in the analyses.
  • The manuscript states that its aim is to develop a landscape attractiveness monitoring framework for the 2008-2018 time period. This implies an evaluation of changes in landscape attractiveness over time. However, the indicators do not capture temporal changes; instead, they come from different points in time during the 2008-2018 time period (i.e., 2012, 2015, and 2018). This is misleading.
  • The spatial resolution of the overall analysis and the individual indicators is not stated. It’s also unclear how differences in spatial resolution were reconciled.
  • Why include BioL and BioV in Table 1 if they were not included in the integration?
  • The methods used for integrating the landscape attractiveness indicators are unclear and not repeatable as written. At this point, I don’t even know what the range of values is for landscape attractiveness. Need much more information.
  • The manuscript notes that “a combination of common knowledge of preferences and relations between landscape characteristics and their functioning” to interpret the landscape indicators with respect to landscape attractiveness. Then various examples are given. For the purposes of full disclosure and repeatability, it would be nice to see references to sources in Table 1 and a brief explanation for each indicator in the text.
  • According to the manuscript, “Identifying trends and patterns in landscape attractiveness for such a dynamic and heterogeneous area is challenging”. However, the dynamic nature of the region appears to be ignored in the analysis, which lumps together data from different single points in time to represent landscape attractiveness somewhere between 2008 and 2018. I recognize that some data are not available for all years and/or that data may not be available for the same year. However, the manuscript doesn’t adequately acknowledge these issues in its current form or justify why the analyses should be considered appropriate despite these issues.
  • The term “geophysical” captures many aspects of the “natural” environment, but not vegetation. Isn’t vegetation is an important and very much visual aspect of landscape attractiveness? Why is this not specifically accounted for?
  • Figure 1/ Table 2 / Text: “Urban fringe” appears to make up much of Flanders. It would be valuable to see this defined in the text and mapped in the figure. Given that it takes up twice the space of urban, it would also make sense to conduct the analysis not just on urban vs. rural, but urban vs. rural vs. urban fringe. If the manuscript stated whether urban fringe was included in the rural or the urban category, I missed it.
  • “total area of protected nature areas increased” – the manuscript does not note multi-temporal data for this item under methods; it doesn’t do so either for other items (e.g., agricultural area)
  • Land cover maps: these were critical in the overall analysis, yet the supplementary materials say nothing about the accuracy of those maps – how was it assessed and what was it?
  • “the protected nature area geodatabase was consulted” – Which? Reference? – same is true for LPIS, the biological valuation map, etc.
  • The manuscript notes that the “aim of this paper is to develop a landscape attractiveness monitoring framework based on geospatial data for Flanders for the 2008-2018 time period”. I’d hoped that the emphasis would be on the development of the landscape attractiveness monitoring framework rather than Flanders. This would make the manuscript more broadly relevant and interesting. Unfortunately, while individual indicators are fairly well described in the supplemental materials, the overall methodology of assigning and integrating attractiveness scores is not well articulated. Some might refer to the paper in its current form as WisCy – wallowing in a specific case.

Minor Issues

  • Geospatial and GIS are related but not the same as implied by “geospatial (GIS)” on p. 1
  • Table 1: note Yes or No for “Included in landscape attractiveness score” for each item, instead of leaving a cell blank or noting “only BioVV is included”
  • “the region was stratified into sub-regions expected to show a homogeneous land cover composition and hence attractiveness” --- this implies that landscape attractiveness is based solely on land cover composition, which it is not, as noted elsewhere in the article; this statement also fails to account for the different land uses that may occur in a single land cover type and presumably influence landscape attractiveness
  • The text defines what is meant by geophysical environment, but the description of the regions focuses mostly on soils, completely ignores climate, and only considers geology and topography to a limited extent. Why?
  • Figure 1. Include the symbology for urban (and rural) in the legend as opposed to the caption; add spatial reference information and north arrow.
  • For international readers, a general reference map showing the location of Flanders within Belgium and perhaps even Europe would be helpful.
  • Figure 6. It is impossible for the human eye to visually distinguish between 15+ shades of green. Recommend using either a continuous color ramp or classifying the scores into not more than 5 or 6 groups. Avoid redundancy in the figure by using a single legend for all scores; consider a 2 x 2 panel for this figure. Add halos to the labels – they are difficult to read on dark green and overlap with lines of the same weight are hard on the eyes.

Summary

The research conducted by the authors has potential to be of broad interest and relevance. Unfortunately, as currently written, the manuscript lacks depth in the literature review and details in the data and methods section. Instead of focusing on the methodological approach, which may be novel and intellectually significant, the paper focuses on specifics regarding Flanders, which is likely of interest to a small audience only. Whether the research was technically sound or not is unclear, because the data and methods were insufficiently described.

Author Response

Dear reviewer,

We appreciate the interesting questions and remarks on our manuscript. We revised the manuscript with care and provide a response to your remarks below.

All the best,

The authors

Major Concerns

  • The manuscript in its current form lacks a deep literature review (only 12 citations in the introduction). As a result, it is unclear how the described work builds on or extends previous efforts and how it is theoretically grounded.

More detail on the applied methodology and references were added to the manuscript. See lines: 61-74; 143-165. Similar to (Sowińska-Świerkosz and Michalik-Śnieżek 2020; Vizzari 2011; Cassatella and Peano 2011), geospatial landscape indicators describing different dimensions of the landscape were used to evaluate landscape attractiveness. The novelty of our work is the evaluation of the studied landscape indicators for different spatial levels (urban vs rural, different agricultural regions) and the integration of the indicators to assess landscape attractiveness.

  • I am not convinced by the argument to “discard the cultural dimension as it can be regarded as 88 static in a time period of ten years, from 2008 to 2018.” Thinking about politics around the world, including the rise of right-wing populism, quite a bit has changed culturally. I’d like to hear a better argument for discarding the cultural dimension or an inclusion of the cultural dimension in the analyses.

We added more detail on this methodological choice to the manuscript see lines 89-90. Historical monuments and historical landscape elements are landscape indicators which are commonly used to characterize the cultural landscape dimension, these can be assumed to be constant from 2008-2018.

  • The manuscript states that its aim is to develop a landscape attractiveness monitoring framework for the 2008-2018 time period. This implies an evaluation of changes in landscape attractiveness over time. However, the indicators do not capture temporal changes; instead, they come from different points in time during the 2008-2018 time period (i.e., 2012, 2015, and 2018). This is misleading.

For indicators where data for multiple years were available changes over time were discussed. For some indicators data from only one point in time were available, therefore we were not able to calculate the changes in time. To avoid confusion we removed/changed sentences throughout the manuscript which could introduce confusion on this matter. In addition, a column indicating the data range for each indicator was added to Table 1.

  • The spatial resolution of the overall analysis and the individual indicators is not stated. It’s also unclear how differences in spatial resolution were reconciled.

The spatial resolution of each database was added to the supplementary materials. All landscape indicators were available for the whole region of Flanders, this allowed us to calculate the indicators for the different scales of analysis i.e. Flanders, urban versus rural, and the seven agricultural regions. The majority of the consulted geodatabases are based on shapefiles. The resolutions of the land cover map (scaled to 30m), boswijzer (10m) and Landsat 8 based NDVI (30m) were deemed suitable to calculate the derived landscape indicators at the studied scales of analysis (i.e. Flanders, urban versus rural, and the seven agricultural regions). The integration of the different indicators was only performed at these scales of analysis, thereby circumventing the issue of different initial resolutions.

  • Why include BioL and BioV in Table 1 if they were not included in the integration?

After reconsideration, we indeed decided to remove BioL and BioV from the manuscript.

  • The methods used for integrating the landscape attractiveness indicators are unclear and not repeatable as written. At this point, I don’t even know what the range of values is for landscape attractiveness. Need much more information.

More detail on the methodology used to integrate the landscape indicators was added to the manuscript. See lines 140-162.

  • The manuscript notes that “a combination of common knowledge of preferences and relations between landscape characteristics and their functioning” to interpret the landscape indicators with respect to landscape attractiveness. Then various examples are given. For the purposes of full disclosure and repeatability, it would be nice to see references to sources in Table 1 and a brief explanation for each indicator in the text.

We added references for each indicator to Table 1 and briefly discussed the interpretation of the landscape indicators in terms of landscape attractiveness it in the method section. See lines 140-162

  • According to the manuscript, “Identifying trends and patterns in landscape attractiveness for such a dynamic and heterogeneous area is challenging”. However, the dynamic nature of the region appears to be ignored in the analysis, which lumps together data from different single points in time to represent landscape attractiveness somewhere between 2008 and 2018. I recognize that some data are not available for all years and/or that data may not be available for the same year. However, the manuscript doesn’t adequately acknowledge these issues in its current form or justify why the analyses should be considered appropriate despite these issues.

We added a column to Table 1 indicating the data range for each studied landscape indicator and eliminated “landscape attractiveness was evaluated for 2008-2018” throughout the text to avoid confusion on this matter.

  • The term “geophysical” captures many aspects of the “natural” environment, but not vegetation. Isn’t vegetation is an important and very much visual aspect of landscape attractiveness? Why is this not specifically accounted for?

Several landscape indicators are included that characterize vegetation (a.o. LCfor, LCnat, BioVV). We decided to evaluate landscape attractiveness for the different agricultural regions since they characterize a particular geophysical environment which determines the type of land use and agricultural activities in these regions, and consequently the landscape. Information on climate in the regions was added, see lines 184-195.

  • Figure 1/ Table 2 / Text: “Urban fringe” appears to make up much of Flanders. It would be valuable to see this defined in the text and mapped in the figure. Given that it takes up twice the space of urban, it would also make sense to conduct the analysis not just on urban vs. rural, but urban vs. rural vs. urban fringe. If the manuscript stated whether urban fringe was included in the rural or the urban category, I missed it.

Different spatial policies are in place for urban and rural areas in Flanders, resulting in differences in landscape appearance, therefore we decided to study landscape attractiveness in rural versus urban areas. The urban fringe is not considered separately in (spatial) policy in Flanders and therefore we did not study landscape attractiveness at this level. As stated by (Pisman et al. 2018) the division of Flanders in rural, urban fringe and urban areas does not aim to be directly applicable for spatial policy. The urban fringe was added to Figure 1 for reference.

  • “total area of protected nature areas increased” – the manuscript does not note multi-temporal data for this item under methods; it doesn’t do so either for other items (e.g., agricultural area)

We added a column to Table 1 indicating the data range for each landscape indicator.

  • Land cover maps: these were critical in the overall analysis, yet the supplementary materials say nothing about the accuracy of those maps – how was it assessed and what was it?

Information on the land cover map is included in the supplementary materials, see lines 564-577.   

  • “the protected nature area geodatabase was consulted” – Which? Reference? – same is true for LPIS, the biological valuation map, etc.

Information on the data source of the consulted geodatabases was added to the manuscript. See lines 101-108

  • The manuscript notes that the “aim of this paper is to develop a landscape attractiveness monitoring framework based on geospatial data for Flanders for the 2008-2018 time period”. I’d hoped that the emphasis would be on the development of the landscape attractiveness monitoring framework rather than Flanders. This would make the manuscript more broadly relevant and interesting. Unfortunately, while individual indicators are fairly well described in the supplemental materials, the overall methodology of assigning and integrating attractiveness scores is not well articulated. Some might refer to the paper in its current form as WisCy – wallowing in a specific case.

We specified the aim of our paper in the introduction section in order to inform the reader on the specific aims of the manuscript.  The applied framework is largely based on the frameworks proposed by (Sowińska-Świerkosz and Michalik-Śnieżek 2020; Vizzari 2011; Cassatella and Peano 2011). Our aim was to demonstrate how such geospatial landscape attractiveness frameworks can be applied in Flanders at different spatial scales and to produce an integrated landscape attractiveness score. We added this explicitly to the research objectives (see lines 64-66). More detail was added on the used methodology to integrate the landscape indicators (see lines 118-162), to ensure that the reader can reproduce the applied methodology.

Minor Issues

  • Geospatial and GIS are related but not the same as implied by “geospatial (GIS)” on p. 1

changed

  • Table 1: note Yes or No for “Included in landscape attractiveness score” for each item, instead of leaving a cell blank or noting “only BioVV is included”

changed

  • “the region was stratified into sub-regions expected to show a homogeneous land cover composition and hence attractiveness” --- this implies that landscape attractiveness is based solely on land cover composition, which it is not, as noted elsewhere in the article; this statement also fails to account for the different land uses that may occur in a single land cover type and presumably influence landscape attractiveness

changed

  • The text defines what is meant by geophysical environment, but the description of the regions focuses mostly on soils, completely ignores climate, and only considers geology and topography to a limited extent. Why?

The regions used in the manuscript are commonly used regions in Flanders. The division of Flanders in the regions is indeed mostly based on soil variability since this is the geophysical variable with the highest variability in Flanders.

  • Figure 1. Include the symbology for urban (and rural) in the legend as opposed to the caption; add spatial reference information and north arrow.

changed

  • For international readers, a general reference map showing the location of Flanders within Belgium and perhaps even Europe would be helpful.

An overview map of Europe was added to Figure 1.

  • Figure 6. It is impossible for the human eye to visually distinguish between 15+ shades of green. Recommend using either a continuous color ramp or classifying the scores into not more than 5 or 6 groups. Avoid redundancy in the figure by using a single legend for all scores; consider a 2 x 2 panel for this figure. Add halos to the labels – they are difficult to read on dark green and overlap with lines of the same weight are hard on the eyes.

The figure was updated

 Summary

The research conducted by the authors has potential to be of broad interest and relevance. Unfortunately, as currently written, the manuscript lacks depth in the literature review and details in the data and methods section. Instead of focusing on the methodological approach, which may be novel and intellectually significant, the paper focuses on specifics regarding Flanders, which is likely of interest to a small audience only. Whether the research was technically sound or not is unclear, because the data and methods were insufficiently described.

References

Cassatella, Claudia, and Attilia Peano, eds. 2011. Landscape Indicators: Assessing and Monitoring Landscape Quality. Springer Netherlands. https://doi.org/10.1007/978-94-007-0366-7.

Pisman, A., S. Vanacker, P. Willems, G. Engelen, and L. Poelmans. 2018. “Ruimterapport Vlaanderen (RURA). Een ruimtelijke analyse van Vlaanderen.” Brussel: departement Omgeving. https://www.vlaanderen.be/publicaties/ruimterapport-vlaanderen-rura-een-ruimtelijke-analyse-van-vlaanderen-2018.

Sowińska-Świerkosz, Barbara, and Malwina Michalik-Śnieżek. 2020. “The Methodology of Landscape Quality (LQ) Indicators Analysis Based on Remote Sensing Data: Polish National Parks Case Study.” Sustainability 12 (April): 2810. https://doi.org/10.3390/su12072810.

Vizzari, Marco. 2011. “Spatial Modelling of Potential Landscape Quality.” Applied Geography, Hazards, 31 (1): 108–18. https://doi.org/10.1016/j.apgeog.2010.03.001.

Round 2

Reviewer 3 Report

I think the authors' idea of attractiveness is clarified by adding the description of the landscape attractiveness based on the literature related to aesthetic attractiveness. The attractiveness looks to rely on the method for evaluating agricultural and ecological landscapes. In addition, a critical review of them would be necessary for the objective evaluation of landscape attractiveness. However, the concept is well organized based on substantial literature, and the analysis has evident originality. Therefore the paper should be published as it is.

Author Response

Thanks for taking the time to review our manuscript.

Reviewer 5 Report

This revised version of the manuscript represents a substantial improvement compared to the original version. I still think the literature review / theoretical grounding and problem statement could be strengthened. I also find that the database integration and spatial resolution issues remain inadequately addressed (e.g., the authors’ response notes, “The integration of the different indicators was only performed at these scales of analysis, thereby circumventing the issue of different initial resolutions” – that means that all raster data had to be aggregated at these scales, yet aggregation procedures are not described, even though they impact results; similarly, it is unclear how final values were derived when boundaries between the vector files and the analysis scales were inconsistent). Moreover, the reliability of the results is difficult to judge, because errors and uncertainties in the input products are not stated (e.g., just because a workflow was elaborate doesn’t mean the final product was accurate – land cover map accuracy was not stated; how it was assessed was not described).

Author Response

To strengthen the theoretical grounding of the applied methodology in the manuscript a paragraph with theoretical background on geospatial derived landscape indicators was added.

Considering your questions on the spatial resolutions and integration procedure: We added information on the mapping scale/accuracy and a reference to the metadata (in the cases that the databases are publicly available) for each dataset that was used as an input in our analysis. In addition, we clarified how the mapped elements were calculated, aggregated and assigned to the studied geographical regions for each scale of analysis for each database.